

# An Improved Geolocation Methodology for Spaceborne Radar and Lidar Systems

Bernat Puigdomènech Treserras[1], Pavlos Kollias[1,2]

[1]Department of Atmospheric and Oceanic Science, McGill University, Montreal, H3A 0B9, QC Canada
[2]School of Marine and Atmospheric Sciences, Stony Brook University, Stony Brook, NY 11790, NY USA

*Correspondence to*: Bernat Puigdomènech Treserras (bernat.puigdomenech-treserras@mcgill.ca)

**Abstract.** Geolocation and co-registration methodologies are essential for the accurate interpretation of observations from spaceborne remote sensors. In preparations for EarthCARE, here, we refine the definition of these techniques and present various examples of geolocation assessments. The geolocation methods build upon earlier work, however, introduces several improvements that have increased the reliability of the geolocation accuracy. The EarthCARE active sensors geolocation methods use coastlines and significant elevation gradients, in both statistical and numerical ways. The effectiveness of the proposed geolocation methods was tested using the extensive record of CloudSat and CALIPSO observations. The EarthCARE active sensors geolocation methods were effective in identifying and correcting a short period of CloudSat observations when the star tracker was not operating properly. In addition, the geolocation methods were able to reproduce the excellent geolocation record of the CloudSat and CALIPSO missions.



## 1 Introduction

The accurate determination of the precise location on Earth's surface and atmosphere that corresponds to a signal received by a spaceborne remote sensing instrument is very important for their interpretation and their synergistic use with signals from other sensors. Geolocation of the signals, and their eventual alignment (co-registration) with datasets from different sensors, are important post-processing methodologies that are vital for the appropriate use of satellite observations. The Earth Cloud Aerosol and Radiation Explorer (EarthCARE), (Wehr et al., 2023), implemented by the European Space Agency (ESA) in cooperation with the Japan Aerospace Exploration Agency (JAXA), stands out as the ESA's most complex Earth Explorer mission. The EarthCARE mission is expected to provide breakthrough observations of aerosols, clouds, precipitation, and radiation, their complex interactions and help improve climate models and weather forecasting (Illingworth et al., 2015).

The EarthCARE satellite payload includes two active sensors, the High-Spectral-Resolution (HSR) Atmospheric LIDAR (ATLID) and the 94-GHz Cloud Profiling Radar (CPR), and two passive sensors, the Multi-Spectral Imager (MSI) and the Broad-Band Radiometer (BBR). While each sensor provides unique measurements capabilities, one of the strengths of the EarthCARE mission is the synergistic use of the multisensor observations. Subsequently, an accurate absolute geolocation and co-registration of all the EarthCARE L1 and L2 products for the interpretation of the information provided by each sensor and the development of synergistic algorithms, like AC-TC (Irbah et al., 2023), ACM-CAP (Mason et al., 2023), or ACM-COM (Cole et al., 2023) is essential. Although the EarthCARE spacecraft Attitude Determination System (ADS) is expected to provide high quality information, the absolute accuracy might be affected by viewing geometry, thermoelastic distortions, or other, yet unidentified, sources (Battaglia and Kollias, 2014). The use of additional techniques is desirable to validate the geolocation information reported in the L1 products and mitigate any unknown effects.

Several methods have already been developed to assess the geolocation of active instruments. Knapp, 2021, demonstrated that the ESA Aeolus wind lidar ground track was directly visible from the Pierre Auger Observatory in Argentina whenever the satellite passed near the facility. The study allowed the geolocation assessment of the lidar dependent on the external observations of the High Elevation Auger Telescope (HEAT). Another example is related to the innovative Doppler capabilities of the EarthCARE CPR that are planned to be used to characterise the CPR off-nadir pointing angle along its orbital track, (Battaglia and Kollias, 2014; Kollias et al. 2023). However, this approach is not sufficient to have a complete and comprehensive view of the instrument geolocation.

Here, the geolocation methodologies for the EarthCARE active instruments specifically focusing on the positions of known natural targets, such as coastlines and significant elevation gradients are presented. These methodologies are based on the earlier contributions (Currey, 2002; Tanelli et al., 2008), designed to be applied to the Cloud-Aerosol Lidar with Orthogonal Polarization (CALIOP) and CPR instruments from the CALIPSO, (Winker et al., 2007) and CloudSat, (Stephens et al., 2002) missions, respectively.



Both techniques are based on the analysis of the instrument's surface returns and are not dependent of external factors. In the case of coastlines, the signal gradient between land and ocean transitions is leveraged to model the coastline signature. Then, 50 through a minimization approach, the absolute geolocation is identified by minimizing the error between a collection of coastline detections and a reference map. Regarding significant elevation gradients, the assessment is performed by comparing the instrument's surface detection height to a reference digital elevation model (DEM). In preparations for the EarthCARE active sensors geolocation and co-registration activity, refined versions of these techniques are presented. In addition, a comprehensive analysis of their performance including actual geolocation errors and lifetime statistics using datasets from 55 both CloudSat and CALIPSO missions is presented.

## 2 Input data

### 2.1 Geospatial reference data

To accurately assess the geolocation and co-registration accuracy of spaceborne lidar and radar instruments, a reliable representation of the Earth's surface is required. Here, the Global Digital Elevation Model (DEM) and Water Bodies Database 60 (WBD) products (Abrams et al., 2020), from the Advanced Spaceborne Thermal Emission and Reflection Radiometer (ASTER) instrument (Abrams et al., 2015), are used. ASTER is a sophisticated 15-channel imaging instrument operated by the National Aeronautics and Space Administration (NASA) and the Japanese Ministry of Economy, Trade, and Industry (METI). Launched on December 1999, aboard NASA's Terra satellite, ASTER is used to create detailed maps of surface temperature of land, emissivity, reflectance, and elevation. Both DEM and WBD products are distributed with a gridding and 65 tile structure of 1 arc second resolution (~30-meter at the Equator) and 1° x 1° tiles with a coverage that spans from 83ºN to 83ºS. The DEM was first released in 2009 and subsequently updated to versions 2 and 3 in 2011 and 2019, respectively. These new releases featured improvements in both horizontal and vertical accuracy and resolution, along with a reduced presence of artifacts. The WBD was created in conjunction with the latest DEM version, providing global coverage of water bodies classified into three categories: oceans, rivers, and lakes, each larger than 0.2 square kilometers. The most recent version of 70 the ASTER products is used in this study. The land to ocean transitions of the WBD are contoured, preserving the original resolution, to enhance the accessibility of coastline information.

### 2.2 Instrument test data

The EarthCARE mission is the follow up to NASA's Afternoon constellation (A-train, Stephens et al., 2018). NASA's A-train featured two active remote sensors, a 94-GHz Cloud Profiling Radar (CPR) on the CloudSat mission (Stephens et al., 2002) 75 and the NASA–Centre National d'Études Spatiales (CNES) Cloud–Aerosol lidar and Infrared Pathfinder Satellite Observations (CALIPSO; Winker et al., 2010).





The CloudSat CPR provides an appropriate source of satellite CPR measurements that can be used to test the EarthCARE CPR geolocation methodology. Both CloudSat and the EarthCARE CPRs operate at the same frequency, 94 GHz. However, while both instruments have similar transceiver design, there are notable differences in their technical capabilities; the EarthCARE

CPR has higher sensitivity (6 dB more sensitive), better vertical sampling (100 versus 240m), higher along-track resolution (500 versus 1100m) and small instantaneous field of view (800 versus 1400m). In addition, the EarthCARE CPR is the first Doppler radar in space for atmospheric applications (Kollias et al., 2014). Here, the CloudSat 2B-GEOPROF data product (Marchand et al. 2008), from June 2$^{nd,}$ 2006, to August 27$^{th}$, 2020, is used.  During this period, only good quality profiles, collected in nominal science mode and in the absence of clouds to prevent attenuation effects, are selected.

Similarly, the CALIPSO mission, (Winker et al., 2007), provides a reliable dataset of CALIOP measurements with high-resolution profiles that can be used to test the geolocation methodology for ATLID. The main differences between the ATLID and CALIOP are the wavelengths (355 nm for ATLID, 532/1064 nm for CALIOP), the footprint (29 versus 90m) and the higher vertical resolution (30 versus 100m). Furthermore, the ATLID is a high spectral resolution lidar (HSRL), while CALIOP is a backscatter lidar. The CALIPSO L1-Standard-V4-51 and L2_333mMLay-Standard-V4-51, NASA/LARC/SD/ASDC,

products collected from June 12$^{th}$, 2006, to June 30$^{th}$, 2023, are used. During this period only profiles collected during nominal science mode and when the Earth's surface is detected are selected.

The decision to rely on CloudSat and CALIOP data is based on its comprehensive coverage and well-established data records. They provide a solid foundation to test the geolocation methods proposed here. However, it's important to acknowledge that incorporating EarthCARE measurements will contribute to reducing uncertainty in the results, given their superior sensitivity,

sampling, and resolution capabilities.

## 3 Geolocation evaluation tool

The geolocation lidar and radar evaluation tool is implemented to detect and quantify the effects of miscalibration and stablish the basis for all the geolocation assessments described in this paper. The tool can be configured with different footprint resolutions and uses the ASTER DEM and WBD products to analyze and simulate the behavior of surface returns over different

regions of interest. An example of the geolocation evaluation tool output is illustrated in Fig. 1.

In the first step, the orbit path is ingested to define the latitude and longitude boundaries of the region to be mapped. Subsequently, DEM tiles and coastal shorelines are extracted from the ASTER dataset (Fig. 1a). To account for the orbit's inclination, the along-track integrated footprint is rotated, and the corresponding portion of the DEM, enclosed within the footprint's extent, is extracted, considering the map resolution. By convolving with the footprint's power distribution function

(Fig. 1b), the tool generates a simulated instrument surface detected height and land/water mask, providing insights into water/land transitions and differences compared to a reference map (Fig. 1c).





Next, the tool accurately identifies the coastline's location relative to the orbit path and returned signal (Fig. 1d). An essential feature of the simulator is its capability to manually introduce artificial along- and cross-track offsets, important for evaluating 110 the geolocation accuracy over different regions of interest.

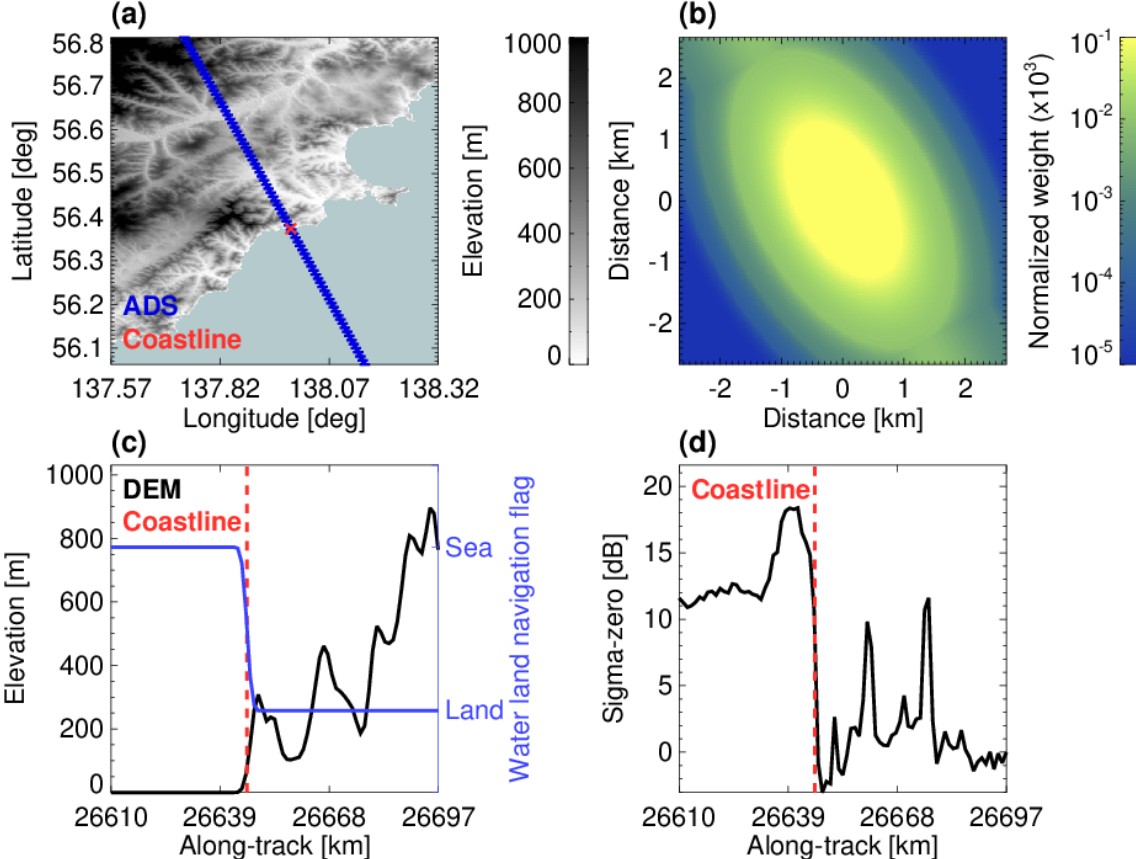

**Figure 1. An example of the geolocation evaluation tool using actual CloudSat data. Panel (a) shows the CloudSat orbit path (blue dots) defined by the Attitude Determination System (ADS) with the identified coastline crossing location (red dot) over the ASTER DEM. Panel (b) shows the CloudSat's along-track integrated footprint rotated according to** 115 **the orbit inclination in ascending mode. Panel (c) shows the simulated DEM using the satellite's footprint (black line), the coastline location (red dashed line) and a simulated water land navigation flag (blue line). Panel (d) shows the coastline location (red dashed line) and the actual measured normalized surface radar cross section ($\sigma_0$) measured by the CloudSat CPR (black line).**



### 3.1 Identification of areas of interest

The geolocation and co-registration techniques presented here are based on the exploitation of Earth' surface signals in areas where significant changes in the measured signals are expected. Coastlines and areas with significant elevation changes are good candidates, however, not all locations are suitable for spaceborne sensor geolocation and co-registration applications. Thus, it is important to identify the optimal locations worldwide that can be effectively used for the analysis.

### 3.1.1 Coastlines

The use of coastlines for spaceborne sensor data geolocation was first introduced for the Earth Radiation Budget Experiment (ERBE) scanner (Hoffman et al., 1987) and it was later refined for the Visible and Infrared Scanner (VIRS) on NASA's Tropical Rainfall Measurement Mission (TRMM, Currey et al., 1998). Here, the version of the coastline algorithm developed for CALIPSO is used (Currey et al., 2002). The coastline detection algorithm analyses the instrument's Earth's surface return along coastline crossings. The pronounced signal gradient between land and ocean transitions is utilized to model the coastline signature and compare it against a reference map. However, not all coastlines are suitable for this kind of analysis. Deserts adjacent to oceans are good candidates, but heterogeneous surface types and irregular terrains can influence the return signal of the instrument in unpredictable ways. Hence, a comprehensive global analysis of all coastlines is necessary to identify the most suitable best coastal regions for geolocating and co-registering the EarthCARE's CPR and ATLID.

Using the CloudSat CPR dataset from 2006 to 2020 a total of 1,079,028 coastline detections are extracted. The orbit of each one of the detections is examined to extract the normalized surface radar cross section ($\sigma_0$) returns that exclusively correspond to ocean and land (Li et al., 2005; Tanelli et al., 2008; Durden et al. 2011). The land and ocean climatological statistics of the CloudSat CPR $\sigma_0$ mean and standard deviation in a gridded map with a resolution of 2° x 2° are shown in Fig. 2.

The results presented in Fig. 2 are consistent with the climatological statistics reported by Durden et al., 2011. Tanelli et al., 2008, identified and quantified additional factors that influence the magnitude of the $\sigma_0$, such as surface winds and sea surface temperature. However, in this study, these factors are not considered since the transition from ocean to land generates a much stronger gradient in $\sigma_0$ than that introduced by changes in near surface winds.

The $\sigma_0$ land and ocean distributions are characterized by their respective mean and standard deviation. To identify the most suitable coastal scenes, the normalized distribution overlapping area between these two distributions is analyzed. The regions with minimal or nearly zero overlapping area are considered potential optimal coastal scene and candidates for the geolocation assessment using coastlines.



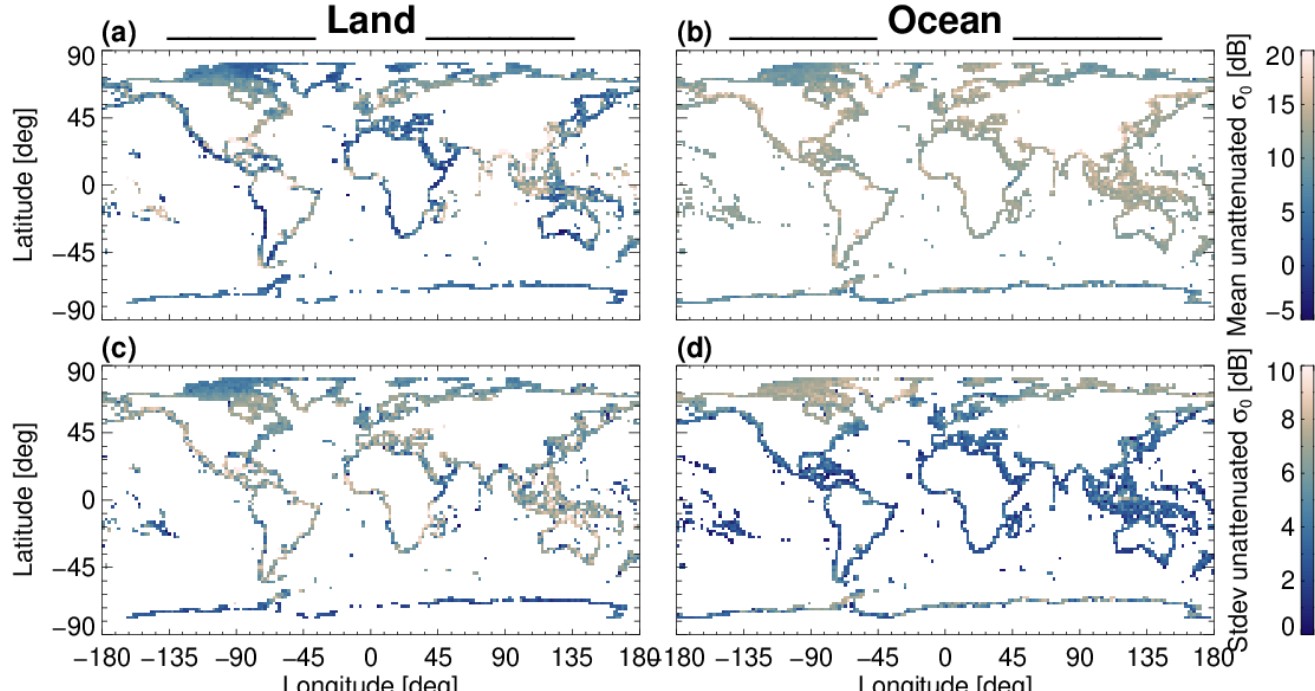

**Figure 2. Climatological CloudSat $\sigma_0$ statistics (mean and standard deviation) over land (panels a and c) and ocean**

**(panels b and d).**

The results of the normalized overlapping areas using the distributions of the 2° x 2° gridded maps shown in Fig. 2, are used to select the most suitable regions for the coastal detection. The best 100 candidates are selected and individually inspected, focusing on the behavior of the $\sigma_0$ measurements from the CloudSat dataset. The suggestion is to identify clear gradients with reduced signal variability in each land and ocean sides. This meticulous assessment and selection of areas of interest

significantly reduces uncertainties in coastline detection, leading to improved overall geolocation and accuracy.

### 3.1.2 Significant elevation gradients

Areas with significant elevation gradients, such as mountains and valleys, offer ideal conditions for geolocation and co-registration studies. They can be used to compare a reference DEM with the instrument's surface detection height. Having steep elevation changes in reduced spatial extents is important for the effectiveness of the technique. If the neighboring areas

have similar heights, the technique will not be able to properly evaluate the geolocation. Additionally, one must consider the instrument's footprint and vertical sampling, as they play a significant role in the analysis; if the elevation gradients within the radar's footprint are lower than the vertical sampling resolution, the technique is also likely to failure. This is particularly important in the case of the CloudSat CPR that has a footprint of 1400m and vertical sampling of 240m.



Tanelli et al., 2008, evaluated the geolocation of the CloudSat CPR using the GTOPO30 DEM data. The use of the coarse resolution DEM (30-arc seconds) led to the conclusion that the CloudSat geolocation is accurate within 500m. The present study benefits from the utilization of the ASTER DEM with a significantly higher resolution, which will help the geolocation analysis.

To find the best scenes with significant elevation gradients, the entire global ASTER dataset is convoluted with the EarthCARE CPR footprint in small domains of 2° x 2° degrees. This convolution enables the computation of both the mean and standard deviation. Subsequently, the scene selection is guided by the DEM standard deviation within the CPR footprint; focusing on identifying domains that exhibit a higher number of standard deviation values surpassing the threshold of 300 meters. Given the higher concentration of mountains in the northern hemisphere (NH), about 89% of selected scenes are located north of the Equator. To ensure a balanced representation and incorporate greater coverage of the southern hemisphere (SH), a minimum inclusion of 30% of areas in the SH is enforced. Figure 3 summarizes the final selected coastal and mountainous areas that will be used for this study and for the EarthCARE CPR and ATLID geolocation and co-registration.

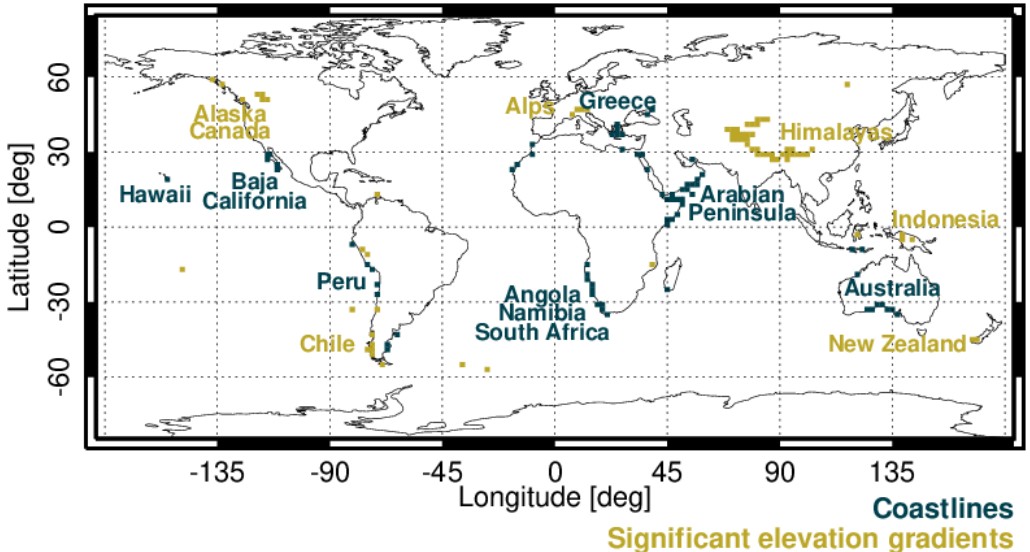

**Figure 3. Selected domain for geolocation studies of coastal scenes and areas characterized by pronounced elevation gradients.**




## 3.2 Geolocation assessment using coastline detection

For lidar instruments, like ATLID and CALIOP, the surface depolarization ratio ($\delta$, Sassen and Zhu, 2009) is the signal that exhibits the most distinct and pronounced gradient between land and water transitions. As a spaceborne lidar beam transects across a coastline, a well-defined step response $\delta$ is produced. The coastline signature is modelled using a cubic fit for at least four contiguous $\delta$ measurements. The inflection point of the fit is the location of the coastline if it falls between the two middle points and the change in signal ($\Delta\delta$) exceeds a predefined threshold set to 0.2.

The coastline detection methodology for the CPR is very similar. However, the inflection point is not expected to accurately represent the actual coastline location. The discrepancy arises from the fact that the $\sigma_0$ values are expressed in decibels (dB). Moreover, the lower along-track resolution of a radar instrument, compared to a lidar, increases complexity. Hence, when utilizing $\sigma_0$ measurements, instead of applying a polynomial fit, the coastline is better detected by interpolating the location of the (locally averaged) $\sigma_0$, computed in linear units, between the ocean and land signatures if the change in signal exceeds a predefined threshold set to 7 dB. Only coastline crossings over the selected areas during clear-sky conditions are used in the geolocation analysis. Figure 4 depicts the geolocation analysis and a summary of the distribution of the $\sigma_0$ and $\delta$ gradients by the CPR and CALIOP, respectively, in 2008.

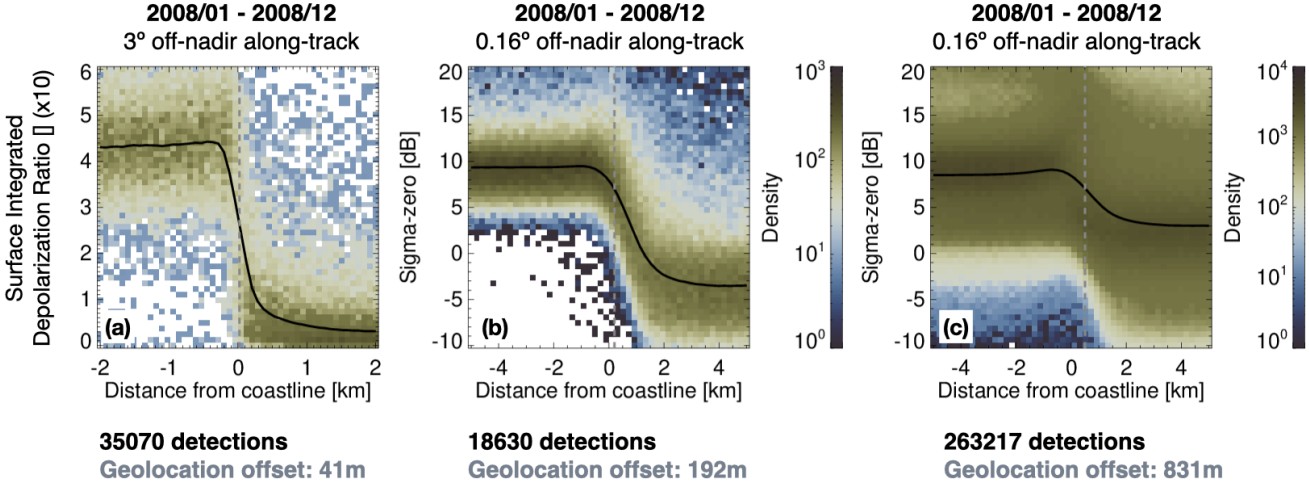

**Figure 4. Histogram of (a) CALIPSO 532nm surface integrated depolarization ratio ($\delta$) and (b) CloudSat normalized radar surface cross section ($\sigma_0$), as a function of distance from the coastline, using measurements from 2008 over the selected coastal areas. Panel (c) presents the CloudSat results when applied to the global coastlines dataset. The black line represents the median of the signal while the grey dashed line indicates the overall coastline detection.**





The CALIPSO geolocation assessment involved 35070 coastline detections and reports a final detection error of 41 m (Fig. 4a). The CloudSat geolocation assessment involved 18630 coastline detections and reports a final detection error of 192 m (Fig. 4b). These findings suggest that the CPR and CALIOP geolocation was very good in 2008. The sharp gradient and stability of the 532 nm surface integrated depolarization ratio, the small footprint of the instrument (90 m) and the high along-track resolution (333 m) contribute to the precision of the CALIPSO results. While the results are less accurate for CloudSat,

the error is less than 10% if the CPR's along-track integrated footprint length of 2.5km is considered. The errors in these results can be considered residual or indicative of accuracy limits. Several factors may contribute to these negligible offsets, such as errors in the reference coastline maps, differences between the modelled and the actual signal processing methods (e.g., asymmetries, resolution variations, etc.) or simply be a consequence of the interaction between the coastline's geometries with respect to the satellite orbital trajectories. Figure 4c depicts the coastline statistics using the global coastlines dataset, without

enforcing any selection criteria on the coastline crossings. In this case, the final detection error is 831 m. The degradation of the geolocation technique (from 192 m to 831 m) indicates that the procedure used for the selection of the coastline crossing is important. The technical characteristics of the EarthCARE active sensors suggest that we should expect improved (locally) surface step responses and thus, hold the potential to improve the coastline geolocation assessments.

    Using a large sample of coastline crossings, this preliminary geolocation analysis, highlights the capabilities of the coastline

detection method and the importance of carefully selecting the coastline locations. However, the analysis does not provide any information about potential along- and cross-track offsets. To address this, the individual detections, generated from orbits with similar orientations, can be grouped for an ensemble analysis in a numerical procedure that minimizes the error between the collection of coastline detections and the reference map.

    For this purpose of identifying the minimum, the simplex method for function minimization (Nelder and Mead, 1965), also

called amoeba or downhill simplex minimization, is an optimal numerical strategy, specifically conceived to find the minimum of an objective function in a multi-dimensional space when the derivatives are unknown. The method uses the complex of a simplex, which is a geometrical figure of N dimensions x N+1 vertices. At each iteration, a cost function is evaluated at each vertex of the simplex and applying a series of transformation, the simplex "slides down" the surface of the function until it finds the minimum. In 2 dimensions (latitude and longitude or along- and cross-track), the simplex is a triangle of 3 vertices

containing the amount to shift an ensemble of coastline crossings. The cost function minimizes the distance between the collection of coastline detections to the digitized map. Examples of this approach are depicted in Fig. 5.



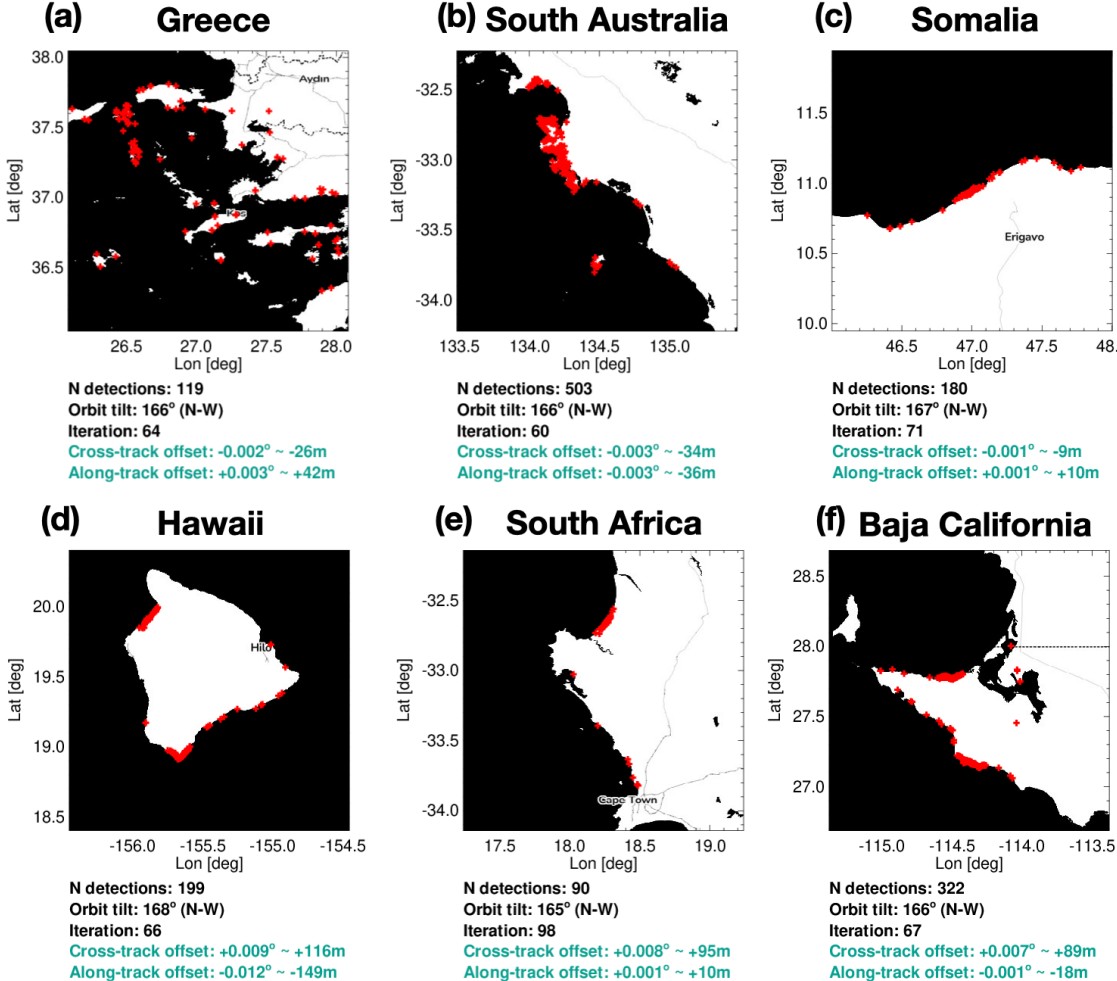

**Figure 5. CloudSat geolocation assessment on some selected coastal scenes. The red dots represent the detections. The base maps are © OpenStreetMap contributors 2015, distributed under the Open Data Commons Open Database License (ODbL) v1.0.**



The results of the amoeba minimization over the coastal scenes are very promising (Fig. 5). Fig. 5a, b, and c represent good cases where the final detected along- and cross-track offsets are consistently below 0.003º. Considering an average satellite
altitude of about 705km, these findings translate to remarkable geolocation errors of less than 50 m. Fig. 5d, e, and f represent cases where higher errors are observed, reaching up to 150 m along-track over the island of Hawaii. The presence of outliers, as the ones visible in Fig. 5f, errors in the reference maps and other factors can influence the minimization and accuracy of the results.

### 3.3 Geolocation assessment using areas with significant elevation gradients

Tanelli et al., 2008, assess the overall accuracy in CPR geolocation by correlating the CPR surface estimated height to the GOTO30 DEM information and concluded that the CPR geolocation is accurate within 500 m. This outcome is mostly due to the resolution of the GOTO30 DEM (approximately 1-km). Here, we benefit from the higher resolution of the ASTER DEM compared to GTOPO30 (1 vs 30 arc-seconds). The technique is applied in post-processing by collecting the instrument's surface detected heights, from orbits with similar orientation, and collocating them across the DEM grid. The simulated
pointing errors, each equivalent to 1 arc-second, are then deliberately introduced in the along- and cross-track directions. The final geolocation error corresponds to the shift that the maximizes the correlation between the instrument and DEM-estimated surface height (convoluted with the footprint's power distribution function).

It is important to consider that in areas with significant elevation gradients, neighboring points often exhibit similar characteristics due to spatial autocorrelation. Given the high resolution of the DEM, relying on only one 'best' solution might
oversimplify the assessment. To strengthen the analysis, points near the maximum correlation are considered. These points can be statistically significant and contribute to the spectrum of possible solutions. To provide a range of geolocation errors, the 95% confidence interval of the 'best' solution is computed using bootstrapping:

1) Pairs of data points from the original dataset (DEM and surface detected heights) are randomly selected and resampled with replacement, creating a thousand of new 'bootstrap' samples. The size of each sample is the same as the original
dataset.

2) For each 'bootstrap' sample, the correlation between the instrument and the DEM-surface elevation estimates is computed.

3) From the distribution of these 'bootstrap' statistics, the 95% confidence interval is determined by selecting the 2.5[th] and 97.5[th] percentiles.



Once the confidence intervals are calculated, the neighboring statistics above the lower 95% confidence interval are considered as additional plausible solutions. Using this approach, the results are reported in probabilistic terms, providing a range of geolocation values, and the method not only prioritize precision (identifying the best offset) but also considers accuracy (acknowledging that nearby points could also be viable solutions). An example of the geolocation assessment using areas with significant elevation gradients is illustrated in Figs. 6 and 7. The geolocation error identified in the area around the Mount

Everest, from January to March 2008, is between 0.001º and 0.012º cross-track and -0.011º and 0.001º along-track with average values of 0.006º and -0.006º, respectively. Considering an average satellite altitude of 705km, these results lead to average mispointing errors of about 73 and -78m with an uncertainty of ±30m. As in the case of coastline detection, the geolocation errors are less than 10% of the CloudSat CPR footprint length.

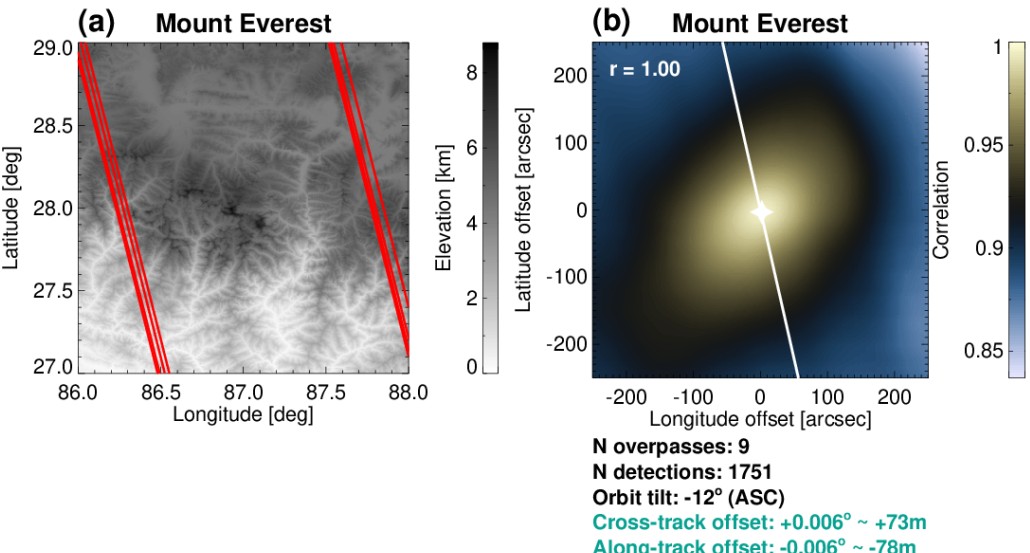

**Figure 6. The CloudSat geolocation assessment around the Mount Everest. Panel a illustrates the area with significant elevation gradients, with red lines representing the CloudSat overpasses from January to March 2008. The statistical**



correlation analysis is depicted in panel b, with the white line representing the satellite path, in ascending orbit, and the filled star denoting the final geolocation offset.

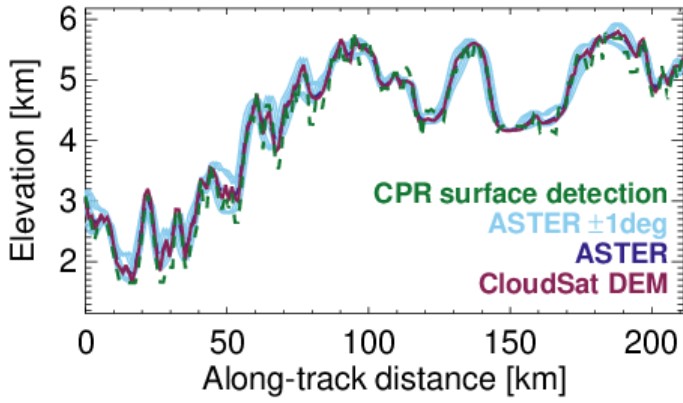


**Figure 7. Comparison between different elevation heights corresponding to the L1 CloudSat surface detection height, the ASTER DEM simulated surface elevations with and without ±1 degree along- and cross-track offsets and the DEM information reported in the CloudSat 2B-GEOPROF product.**





### 3.4 Combined geolocation statistics

A sensitivity study indicated that a three-month period is needed to accumulate enough overpasses over each scene to conduct the geolocation assessment using the coastlines approach. Shorter time periods can be used with surface detections. Irrespectively of the features used, coastlines or terrain, the use of a longer data records will always improve the robustness of the statistics. Ultimately, it is important to balance the number of detections with the desirable temporal resolution of the geolocation assessment. Furthermore, it is anticipated that the confidence levels of results will vary across scenes, influenced by several factors, including: the number of detections within each scene, the accuracy of the reference maps and the convergence and residual errors of each individual analysis.

The number of detections, function and residual errors will be known for each statistical analysis and this information can be used in the final aggregated statistics. The scenes with a higher number of samples are given a higher weight in the average. To refine the results and balance detection numbers with accuracy, the function and residual errors can be leveraged in the detection of coastlines to filter out outliers or cases where convergence is not reached. This approach needs to be performed carefully because small errors are not always indicative of good results. Particularly, straight coastlines have the potential to bias the statistics by reporting inaccurate geolocation results with small residual errors. Most of the times, the results need to be manually reviewed to prevent such situations. Finally, since the results from the significant elevation gradients technique are presented through confidence intervals, they can be aggregated to report the final geolocation error in probabilistic terms. Given the higher count of surface height detections compared to coastline detections within a 2° x 2° domain, this method is anticipated to yield the most accurate results.

### 4. Case study: A CloudSat CPR period with geolocation issues

CloudSat was launched from the Vandenberg Air Force Base, in California, on April 28, 2006. Although the mission was expected to have a two-year life, it exceeded expectations and continued to provide valuable data for many years beyond that. In 2011, the satellite started operating in daytime-only operations due to a battery malfunction, requiring sunlight to power the radar (Witkowski et al., 2018). The satellite encountered further difficulties with a reaction wheel failure in 2018 that forced the satellite to exit the A-Train (B. M. Braun et al., 2019), followed by another failure in one of the remaining wheels in 2020. Finally, the CloudSat ceased operations on December 20, 2023, concluding a prolific 17-year and 8-month legacy of scientific observations. Throughout the CloudSat mission, the ADS used a star tracker to properly estimate the positioning of the CPR antenna. In late July 2019, the ADS started experiencing operational issues, which translated into geolocation errors. While the exact cause of these errors has not yet been determined, it is believed that an unexpected inclination in the satellite's platform might have compromised the accuracy of the star tracker. The problems are suspected to be caused by the software and, more precisely, the star catalogues internally used (NASA's Jet Propulsion Laboratory (JPL) personal communication).



This period presented a very valuable opportunity to test and apply the geolocation techniques presented here, utilizing a unique satellite dataset that includes actual, not fully characterized CPR antenna mispointing. After collecting the data of the specified period, the geolocation assessment is initially performed using both the minimization approach on coastline detections and the significant elevation gradients in consecutive periods of 3 months. Examples of the geolocation analysis are shown in Figs. 8 and 9.

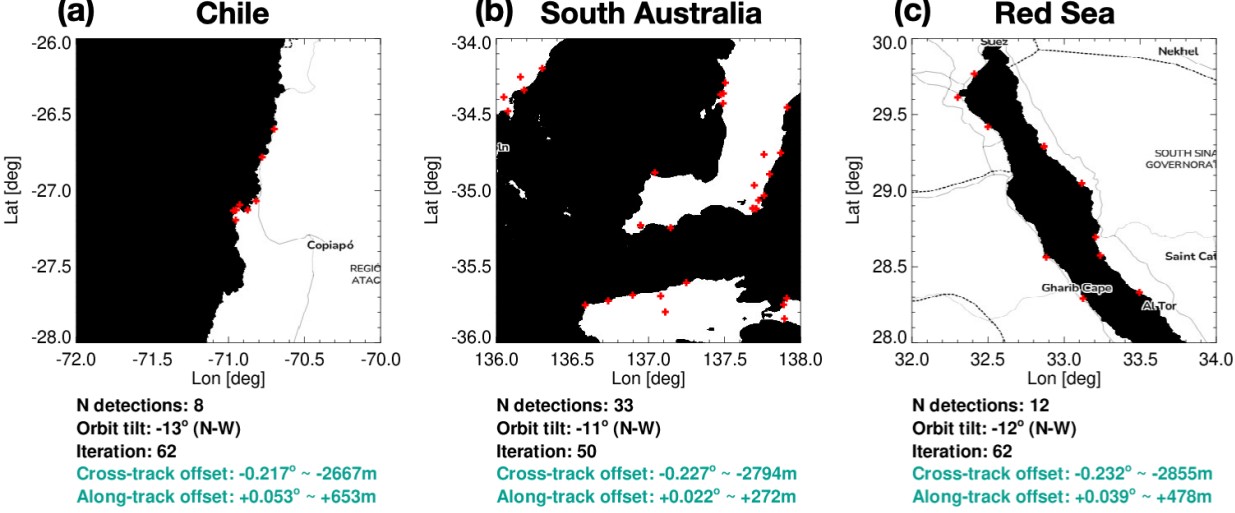

315

**Figure 8. Examples of the coastline geolocation assessment using CloudSat data from July to September 2020. The red dots represent the detections. The base maps are © OpenStreetMap contributors 2015, distributed under the Open Data Commons Open Database License (ODbL) v1.0.**

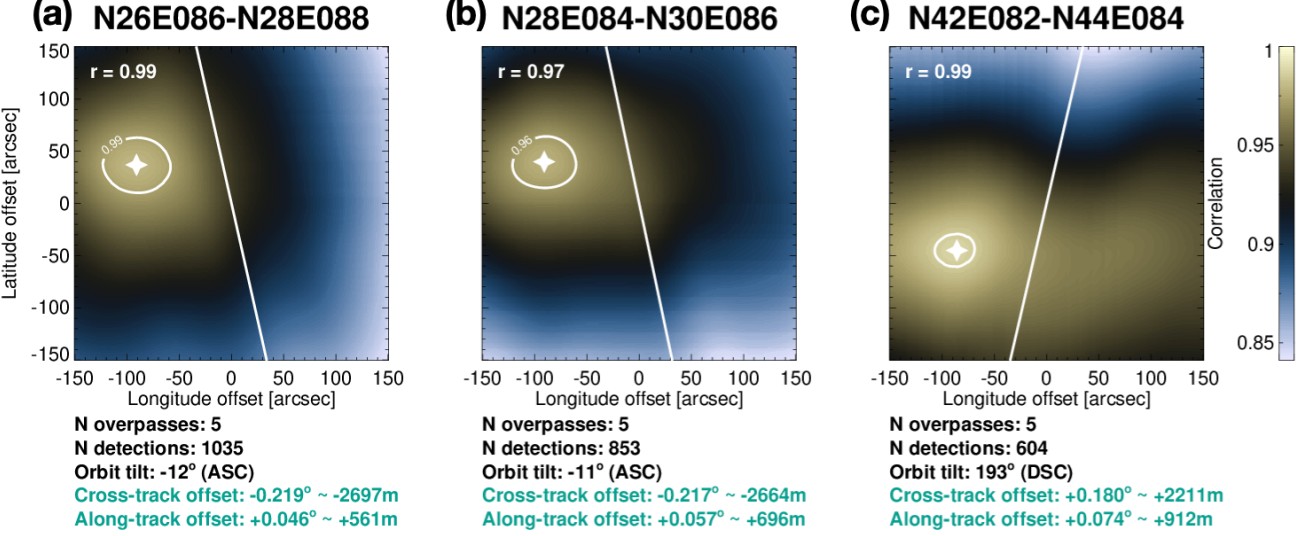





**Figure 9. Examples of the geolocation error identification using the CloudSat data from July to September 2020, two ascending orbits and one descending over areas with significant elevation gradients. The solid white line represents the satellite path, the maximum correlation is indicated with a filled star and the circle around it corresponds to the 95% confidence interval.**

The three examples of coastline detections illustrated in Fig. 8, located over the coasts of Chile, South Australia, and the Red Sea, exhibit similar geolocation offsets ranging from -0.217 to -0.232º cross-track and from +0.022º to +0.053º along-track. The reported along- and cross-track offsets are much higher than those determined in other periods of the CloudSat operational record, thus, confirming the presence of a non-characterized CPR antenna mispointing.

The three examples of significant elevation gradients detections shown in Fig. 9 are located over the Himalayas. Panels a and b refer to ascending orbits and reveal very similar geolocation offsets ranging from -0.217º to -0.219 cross-track and from +0.046º to 0.057º along-track. Panel c illustrates a descending orbit, and it is worth noticing that the cross-track errors have inverse sign. While the exact reason of these sign differences in cross-track offsets between ascending and descending orbits is unknown, it is assumed that the discrepancy is caused by the star tracker software not properly accounting for the rotation of the platform.

The coastline and rough terrain analyses yield similar results. With data from 70 coastlines and 70 rough terrain scenes (divided in groups of similar orbit inclinations), the results are combined to provide a comprehensive statistical assessment of the CloudSat CPR antenna mispointing (Fig. 10).

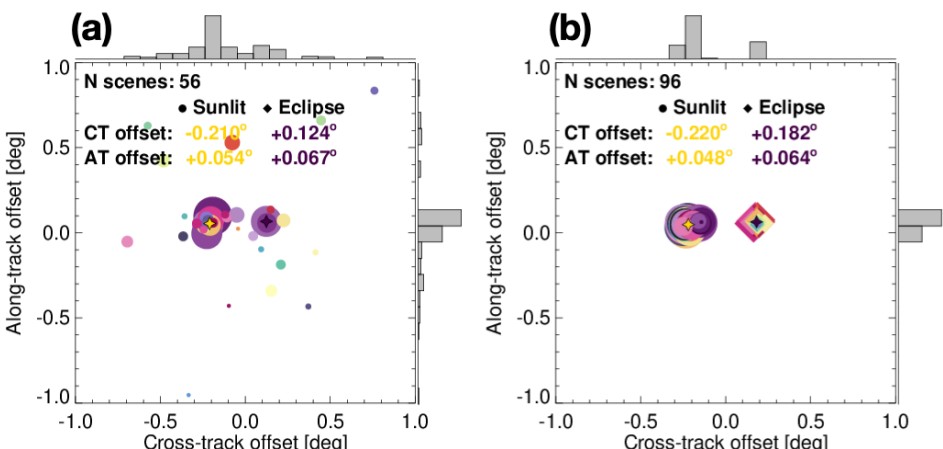

**Figure 10. Combined geolocation statistics of the (a) coastline and (b) significant elevation scenes, using the CloudSat data from July to September 2020. The size of each symbol is indicative of the number of overpasses. Circles and diamonds represent ascending and descending orbits, respectively. A distinctive color is used to identify the scene and filled stars denote the average, yellow for ascending and purple for descending.**



The results shown in Fig. 10 are computed using a weighted average taking the number of detections into account. Final values are similar for both techniques and suggest that the errors in the CloudSat's star tracker introduced a geolocation error between
-0.210 to -0.215º cross-track and 0.048º to 0.054º along-track, during the period from July to September 2020. The coastlines results exhibit more variability compared to the significant elevation gradients. This was already expected, considering the differences between number of samples and resolutions employed by each method. Specifically, the coastline technique often detects only one or a few coastline crossings within each 2° x 2° domain, with an along-track resolution of 1.1 km. On the other hand, the elevation gradients technique benefits from numerous surface detections within each domain and leverages the
advantage of the vertical sampling resolution of 240 meters.

After several iterations, and with the help of the results presented here, JPL identified the error in the star tracker catalogues (JPL personal communication). Following the correction update, the CloudSat geolocation finally reached an acceptable level of accuracy. The final along- and cross-track median geolocation errors for ascending orbits are -0.006º and 0.025º, respectively, corresponding to approximately 75 and 245m.

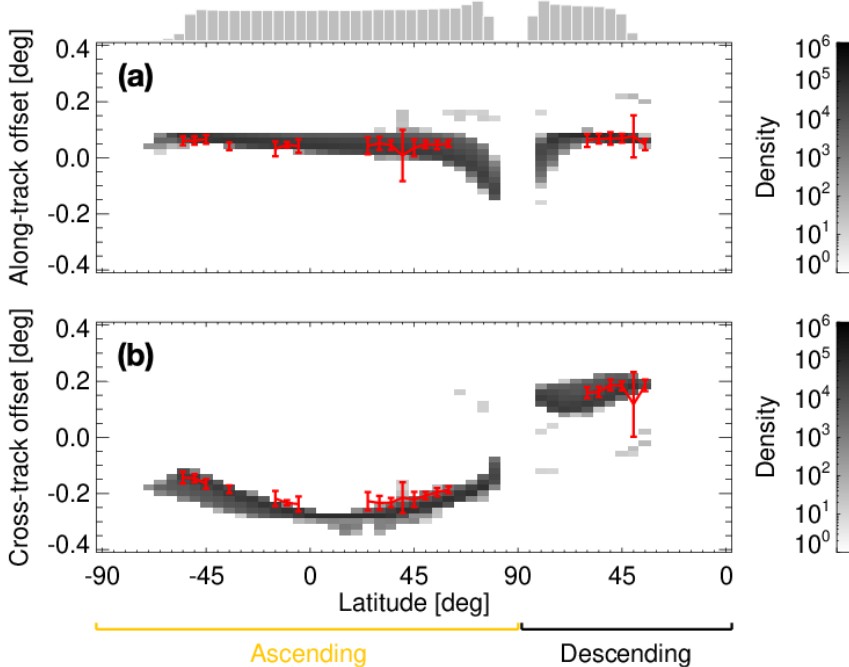


**Figure 11. (a) along- and (b) cross-track validation offsets. The background histogram illustrates the validation geolocation offsets, based on the latitude/longitude differences between the two CloudSat datasets — before and after the star tracker catalogue correction — covering data from July to September 2020. The red lines represent the 95% confidence intervals of the results obtained from the geolocation tools.**






To further evaluate the findings of the geolocation methods, a final analysis is conducted: the datasets from before and after the catalogue correction are inter-compared. To accomplish this, the latitude and longitude differences between the two datasets are translated into along- and cross-track offsets. Figure 11 illustrates these differences as a function of latitude. The correspondence illustrated in Fig. 11, between the geolocation and validation offsets is remarkably good. There is a small bias

in the cross-track figure, panel b, due to the residual mispointing that the star tracker correction failed to completely fix. The inverse sign between the ascending and descending phases identified before, is also validated here. Interestingly, the geolocation techniques can detect the harmonic-type mispointing behavior. Due to the selection of scenes being distributed across both the north and south hemispheres, the results can effectively be illustrated as a function of latitude.

## 5. CloudSat and CALIPSO lifetime geolocation statistics

The methods described here are applied to the entire CloudSat and CALIPSO data records to provide lifetime geolocation statistics for the spaceborne radar and lidar instruments (Fig. 12 and 13).

The CloudSat geolocation statistics, calculated from three-month data segments spanning from 2006 to 2019, show consistent along- and cross-track offsets of 0.002º and 0.0011º, respectively, representing a small fraction of the total CloudSat footprint length. The only exception is identified between the months of July 2015 to March 2016 where the geolocation error increases

in both along- and cross-track directions. These problems were identified and corrected through a reaction wheel rebalancing on January 22$^{nd}$ and February 10$^{th}$, 2016 (BAE Systems, Inc. personal communication).

The CALIPSO geolocation statistics, calculated from three-month data segments spanning from 2006 to 2023, show very consistent and stable along- and cross-track offsets of -0.011º and -0.005º, respectively. Beginning November 28$^{th}$, 2007, the CALIPSO off-nadir angle was permanently changed to 3.0 degrees and the along-track geolocation error increased at that

same time.





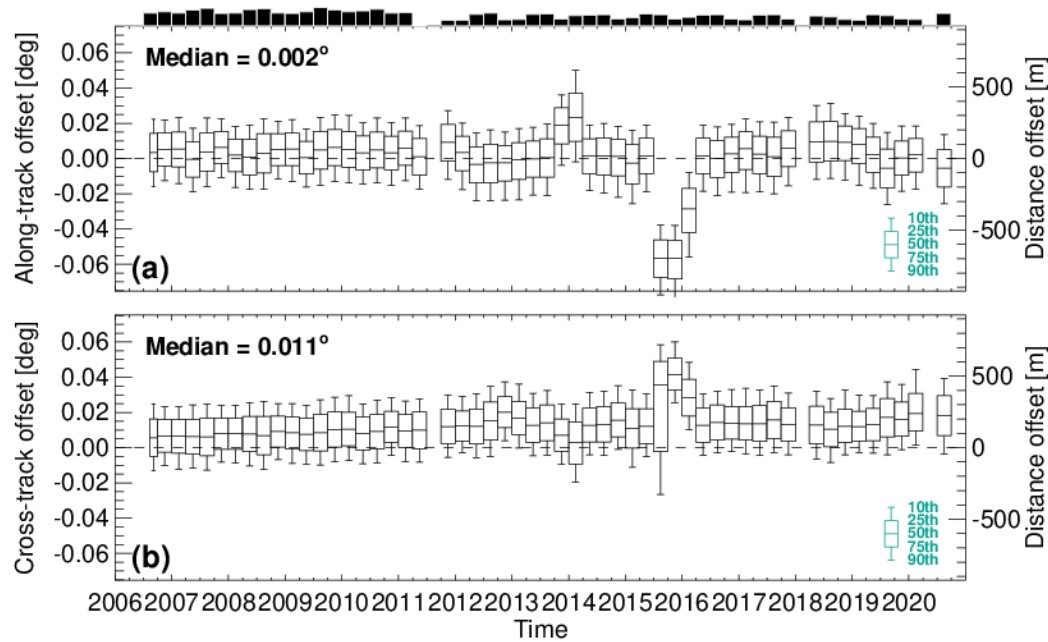

**Figure 12. (a) along-track and (b) cross-track CloudSat lifetime geolocation statistics.**

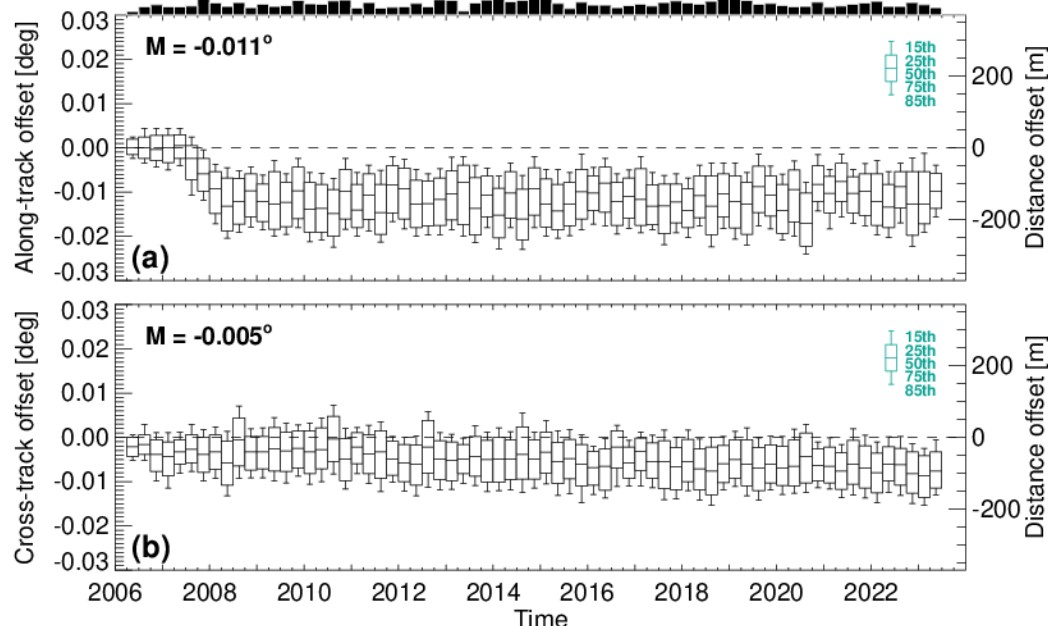

**Figure 13. (a) along-track and (b) cross-track CALIPSO lifetime geolocation statistics.**



## 6. Summary

The joint ESA/JAXA EarthCARE mission features the first Cloud Profiling Radar (CPR) with Doppler capability. The Doppler capability of the CPR is expected to provide unique global observations of convective vertical air motion in shallow and deep convection and information of hydrometeors size and density. In addition, the 355nm-high spectral resolution lidar (HSRL) is
expected to provide improve characterization of ice habits and improve aerosol properties and typing. Along with the passive sensors, the EarthCARE satellite mission is expected to provide a global dataset of cloud, aerosol, and radiation properties.

The accurate determination of the precise location on Earth's surface and atmosphere that corresponds to a signal received by a spaceborne remote sensing instrument is very important for their interpretation and their synergistic use with signals from other sensors. The critical importance of accurate geolocation and co-registration requires comprehensive testing, to ensure
that the methodologies will properly work when EarthCARE is in orbit.

Here, the geolocation methods for the EarthCARE active sensors were presented. The geolocation methods build upon earlier work, however, introduce several improvements that have increased the reliability of the geolocation accuracy. The EarthCARE active sensors geolocation methods use coastlines and significant elevation gradients, in a statistical and numerical way. The effectiveness of the proposed geolocation methods was tested using the extensive record of CloudSat and CALIPSO
observations. The EarthCARE active sensors geolocation methods were effective in identifying and correcting a short period of CloudSat observations when the star tracker was not operating properly. In addition, the geolocation methods were able to reproduce the excellent geolocation record of the CloudSat and CALIPSO missions.

The co-registration is another essential requirement when datasets from different instruments need to be combined. In the EarthCARE mission, this will be the case for several synergistic algorithms that utilize both radar and lidar measurements, like
AC-TC, ACM-CAP, and ACM-COM. Hence, a valid co-registration between the EarthCARE ATLID and CPR is very important. The absolute geolocation techniques described here will be applied to both instruments and the co-registration will be built on the statistical comparison between each individual assessment. Apart from that, the co-registration of the CPR and ATLID will also be performed in the along-track direction, using cross-correlation of the surface height detection over the selected areas with significant elevation gradients. Unfortunately, this kind of analysis could not easily be implemented to co-
registrate CloudSat and CALIPSO. The instruments are placed on different platforms and, even though they follow similar orbits, do not perfectly trace identical paths.

## Acknowledgments

Special thanks to Matthew Lebsock and Gregg Dobrowalski from NASA's Jet Propulsion Laborator, and Heidi Hallowell from BAE Systems, Inc., for providing access to the CloudSat data after correcting the geolocation issues, sharing the attitude target
log files, and discussing the results presented here.



**Data availability**

The CloudSat case with geolocation issues can be downloaded from the official CloudSat DPC web site (www.cloudsat.cira.colostate.edu), release version 5. The corrected version is not yet available to the public at the time of submitting this manuscript, but it will be in the next few months as release version 6 (JPL personal communication).


**Author contributions**

BPT develop the geolocation tools for the EarthCARE active sensors, performed the analysis and draft an early version of the manuscript. PK contributed to the evaluation of the results and the writing of the manuscript.

**Competing interests**

One of the authors is a member of the editorial board of Atmospheric Measurement Techniques.

**Disclaimer**

Publisher's note: Copernicus Publications remains neutral regarding jurisdictional claims in published maps and institutional affiliations.

**Special issue statement**

This article is part of the special issue "EarthCARE Level 2 algorithms and data products". It is not associated with a conference.

**Financial support**

Work done by Bernat Puidgomènech Treserras and Pavlos Kollias was supported by the European Space Agency (ESA) under
the Clouds, Aerosol, Radiation – Development of INtegrated ALgorithms (CARDINAL) project (grant no. RFQ/3-17010/20/NL/AD).



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
