# Peer review of "An Improved Geolocation Methodology for Spaceborne Radar and Lidar Systems"

_EGUsphere, 2024_

## Referee Comment (RC2)

General Comments:

Overall, the paper is technically sound and shows how the geo-location and co-registration approaches used for the upcoming EarthCARE mission are mature when compared with other flight missions. Of particular note are the authors interaction with the project CloudSat to understand and guide their analysis. Several technical comments have been provided, but presented to clarify and not take away from the contents of the paper.

Specific Comments:

1) Can you please expand on what you mean by 'when the Earth's surface is detected' in Section 2.2? Are you also only using clear air instances like you did for CPR when you say 'absence of clouds to prevent attenuation affects', or using those profiles when the surface is seen (i.e. signal hasn't been attenuated).

   *The CALIPSO L1-Standard-V4-51 and L2_333mMLay-Standard-V4-51, NASA/LARC/SD/ASDC, products collected from June 12th, 2006, to June 30th, 2023, are used. During this period only profiles collected during nominal science mode and when the Earth's surface is detected are selected.*

2) Is the Geolocation Evaluation Tool a software package that will be made publicly available?

3) In section 3.1.2 was there a specific reason why a threshold of 300m was used to identify those scenes with elevated gradients (some x number of degrees from expected footprint)? Or was this determined as part of the assessment described in section 3.1.1?

4) In section 3.2 I don't understand what is meant when you say 'The discrepancy arises from the fact that the values are expressed in decibels (dB)' when describing problems using the CPR? Can you clarify what is meant by that sentence? Is this addressed when later in the paragraph you mention conversion of this parameter into 'linear units'?

5) Why was 2008 selected as your analysis year for section 3.2?

6) As you note later in the paper (section 5, Figure 13) CALIPSO changed its off nadir-angle from 0.3 to 3.0 degrees in 2007. Presumably, and if I understand correctly how the error analysis was done for that section, the error increased at 3.0. So, would the detection error of 41 m improve using 0.3 degrees, and is that something to note in section 3.2, or is it not relevant?

7) In section 3.3 you note that Tanelli et al. 2008 compared the CPR with GTOPO30 using their R04 release. Based on this analysis they updated their DEM for R05 to be a combination of multiple sources, including maps from GTOPO30, ASTER GDEM, Greenland (Griggs and Bamber), Antartica (DiMarzio), and STRM. Given you used R05 for your analysis I recommend that you note this change. Additionally, based on this CloudSat analysis CALIPSO changed their DEM from GTOPO30 to this blended CloudSat DEM, and again as you use V4.51 CALIPSO data which employs this new DEM I recommend you mention that.

8) Recommend re-wording the last sentence of section 5 to indicate that the off-nadir angle at the start of CALIPSO science operations (June 13, 2006) was at 0.3 and was permanently changed to 3.0 to account for observed specular reflection due to ice clouds.

**Technical Corrections:**

1) There is at least one instance in the paper (first paragraph in section 2.2, note below) in which there is a clear distinction between Satellite (CloudSat) and Instrument (CPR) but not followed for CALIPSO; CALIPSO satellite, CALIOP instrument. You've correctly done this in other sections but recommend being consistent throughout to avoid any potential confusion.

   *The EarthCARE mission is the follow up to NASA's Afternoon constellation (A-train, Stephens et al., 2018). NASA's A-train featured two active remote sensors, a 94-GHz Cloud Profiling Radar (CPR) on the CloudSat mission (Stephens et al., 2002) 75 and the NASA–Centre National d'Études Spatiales (CNES) Cloud–Aerosol lidar and Infrared Pathfinder Satellite Observations (CALIPSO; Winker et al., 2010).*

2) In section 2.2 the way that you differentiate the vertical resolution between ATLID and CALIOP could be misinterpreted. When describing the difference in wavelength and footprint you put ATLID first (355 nm and 29 m), but vertical resolution you put CALIOP first (30 m). Recommendation for clarity is to say, '*and the higher vertical resolution (100 versus 30)*'. OR you could also put ATLID and CALIOP in each qualifier and say, '*the footprint (29 m for ATLID, 90 m for CALIOP) and the higher vertical resolution (30 m for CALIOP, 100 m for ATLID).*'

   *The main differences between the ATLID and CALIOP are the wavelengths (355 nm for ATLID, 532/1064 nm for CALIOP), the footprint (29 versus 90m) and the higher vertical resolution (30 versus 100m).*

3) To continue on point #2 above, vertical resolution of 30 m for CALIOP is only for the 532 nm channel and only goes from -0.5 to 8.3. 1064 nm uses a 60 m resolution at this low altitude. You may want to designate that the 30 m is only for the 532 nm channel.

4) The DOI citation for the CALIPSO 333m Merged Layer product is incorrect.

   NASA/LARC/SD/ASDC.: CALIPSO Lidar Level 2 1/3 km Merged Layer, V4-51. NASA Langley Atmospheric Science Data Center DAAC, doi: , 2022.

   Needs to be -> https://doi.org/10.5067/CALIOP/CALIPSO/CAL_LID_L2_333mMLay-Standard-V4-51

---

## Author Comment (AC1)

**General Comments:**
Overall, the paper is technically sound and shows how the geo-location and co-registration approaches used for the upcoming EarthCARE mission are mature when compared with other flight missions. Of particular note are the authors interaction with the project CloudSat to understand and guide their analysis. Several technical comments have been provided, but presented to clarify and not take away from the contents of the paper.

We would like to thank the reviewer for taking the time to read the manuscript and provide useful comments that will help improve the clarity and overall quality of our work.

**Specific Comments:**
1) Can you please expand on what you mean by 'when the Earth's surface is detected' in Section 2.2? Are you also only using clear air instances like you did for CPR when you say 'absence of clouds to prevent attenuation affects', or using those profiles when the surface is seen (i.e. signal hasn't been attenuated).

*The CALIPSO L1-Standard-V4-51 and L2_333mMLay-Standard-V4-51, NASA/LARC/SD/ASDC, products collected from June 12th, 2006, to June 30th, 2023, are used. During this period only profiles collected during nominal science mode and when the Earth's surface is detected are selected.*

The CALIOP backscatter signal from the surface is typically much stronger than the atmospheric returns, which allows for surface detection. The surface is detected when the signal has not been completely attenuated and the surface can be 'seen'. Since LIDAR signals are easily attenuated, we simplified our approach by assuming that when the surface is detected, there is no significant attenuation in the profile, so we didn't filter the profiles by the presence of clouds as in the case of the CPR. Our analyses demonstrated that this assumption did not compromise the results.

We have rephrased the sentence for clarity:
"During this period, only profiles collected during nominal science mode and when the Earth's surface is detected —indicating that the signal has not been completely attenuated and the surface is seen— are selected. "

2) Is the Geolocation Evaluation Tool a software package that will be made publicly available?

No, there have been no requests from ESA to make the software package available. Due to the need for several datasets and various processing routines, we have decided not to make it publicly accessible. However, the C-APC processor (Kollias et al., 2023) is licensed under the Apache License Version 2 and the codes will likely be distributed publicly in the future. The C-APC algorithm is referenced in the paper and uses natural targets to apply any necessary corrections to the CPR raw Doppler velocities due to the CPR antenna pointing.

3) In section 3.1.2 was there a specific reason why a threshold of 300m was used to identify those scenes with elevated gradients (some x number of degrees from expected footprint)? Or was this determined as part of the assessment described in section 3.1.1?

Good point. Yes, the CloudSat CPR vertical sampling is 240 meters, so we chose a threshold slightly above that value. We have clarified this in the text.

4) In section 3.2 I don't understand what is meant when you say 'The discrepancy arises from the fact that the values are expressed in decibels (dB)' when describing problems using the CPR? Can you clarify what is meant by that sentence? Is this addressed when later in the paragraph you mention conversion of this parameter into 'linear units'?

Since $\sigma_0$ is expressed in dB, the inflection point of a fitted polynomial would not accurately represent the land/water transition. This is analogous to comparing the average of two values in linear versus logarithmic units — it is not the same. While the polynomial fit works when $\sigma_0$ is expressed in linear units, the transition is better detected through interpolation. All this information is thoroughly addressed in the text.

5) Why was 2008 selected as your analysis year for section 3.2?

It was an arbitrary selection of one of the golden years (when the instruments didn't have major problems). The rest of the years are presented in section 5.

6) As you note later in the paper (section 5, Figure 13) CALIPSO changed its off nadir-angle from 0.3 to 3.0 degrees in 2007. Presumably, and if I understand correctly how the error analysis was done for that section, the error increased at 3.0. So, would the detection error of 41 m improve using 0.3 degrees, and is that something to note in section 3.2, or is it not relevant?

Thanks for this comment. You are right in noticing that the change in off nadir-angle indeed increased the geolocation error and yes, the errors are smaller when the instrument was pointing at 0.3 degrees in 2006 and 2007. We have introduced your suggestion in the text in section 3.2

"It is worth noting that these geolocation results are even better in previous years, as discussed in Section 5"

7) In section 3.3 you note that Tanelli et al. 2008 compared the CPR with GTOPO30 using their R04 release. Based on this analysis they updated their DEM for R05 to be a combination of multiple sources, including maps from GTOPO30, ASTER GDEM, Greenland (Griggs and Bamber), Antartica (DiMarzio), and STRM. Given you used R05 for your analysis I recommend that you note this change. Additionally, based on this CloudSat analysis CALIPSO changed their DEM from GTOPO30 to this blended CloudSat DEM, and again as you use V4.51 CALIPSO data which employs this new DEM I recommend you mention that.

Thanks for this comment, the information has been added to the text:

"Tanelli et al., 2008, evaluated the geolocation of the CloudSat CPR using the GTOPO30 DEM data. The use of the coarse resolution DEM (30-arc seconds) led to the conclusion that the CloudSat geolocation is accurate within 500m. Based on these findings, the DEM was updated in the R05 release to a blend of multiple sources, including maps from GTOPO30, ASTER, Greenland (Bamber et al., 2013), Antarctica (Dimarzio, J. 2007), and SRTM (NASA, 2013). In parallel, the CALIPSO DEM was also updated to this blended DEM. In this study, we leverage the ASTER to enhance the accuracy of our geolocation analysis when compared to the study by Tanelli et al., 2008."

8) Recommend re-wording the last sentence of section 5 to indicate that the off-nadir angle at the start of CALIPSO science operations (June 13, 2006) was at 0.3 and was permanently changed to 3.0 to account for observed specular reflection due to ice clouds.

Thank you very much for this comment, it has been added to the text:

"At the start of CALIPSO science operations on June, 2006, the off-nadir angle was set at 0.3 degrees. However, to account for observed specular reflection due to ice clouds, the off-nadir angle was permanently changed to 3.0 degrees on November 28th, 2007, resulting in an increase in along-track geolocation error at the same time."

**Technical Corrections:**
1) There is at least one instance in the paper (first paragraph in section 2.2, note below) in which there is a clear distinction between Satellite (CloudSat) and Instrument (CPR) but not followed for CALIPSO; CALIPSO satellite, CALIOP instrument. You've correctly done this in other sections but recommend being consistent throughout to avoid any potential confusion.

*The EarthCARE mission is the follow up to NASA's Afternoon constellation (A-train, Stephens et al., 2018). NASA's A-train featured two active remote sensors, a 94-GHz Cloud Profiling Radar (CPR) on the CloudSat mission (Stephens et al., 2002) 75 and the NASA–Centre National d'Études Spatiales (CNES) Cloud–Aerosol lidar and Infrared Pathfinder Satellite Observations (CALIPSO; Winker et al., 2010).*

Thanks for noticing this, all the instances thorough the manuscript have been corrected for consistency.

2) In section 2.2 the way that you differentiate the vertical resolution between ATLID and CALIOP could be misinterpreted. When describing the difference in wavelength and footprint you put ATLID first (355 nm and 29 m), but vertical resolution you put CALIOP first (30 m). Recommendation for clarity is to say, '*and the higher vertical resolution (100 versus 30)*'. OR you could also put ATLID and CALIOP in each qualifier and say, '*the footprint (29 m for ATLID, 90 m for CALIOP) and the higher vertical resolution (30 m for CALIOP, 100 m for ATLID).*'

*The main differences between the ATLID and CALIOP are the wavelengths (355 nm for ATLID, 532/1064 nm for CALIOP), the footprint (29 versus 90m) and the higher vertical resolution (30 versus 100m).*

Thanks for noticing this error, it has been addressed.

3) To continue on point #2 above, vertical resolution of 30 m for CALIOP is only for the 532 nm channel and only goes from -0.5 to 8.3. 1064 nm uses a 60 m resolution at this low altitude. You may want to designate that the 30 m is only for the 532 nm channel.

(100m for ATLID, 30m for the 532 nm channel covering altitudes between -0.5 km and 8.3 km, and 60m for the 1064 nm channel at the same low altitudes for CALIOP).

1.  4) The DOI citation for the CALIPSO 333m Merged Layer product is incorrect.

NASA/LARC/SD/ASDC.: CALIPSO Lidar Level 2 1/3 km Merged Layer, V4-51. NASA Langley Atmospheric Science Data Center DAAC, doi: 10.5067/CALIOP/CALIPSO/CAL_LID_L2_333mMLay-Standard-V4-5, 2022.

Needs to be -> https://doi.org/10.5067/CALIOP/CALIPSO/CAL_LID_L2_333mMLay-Standard-V4-51

Thanks, it has been corrected!

---

## Author Comment (AC2)

Reviewer 1

This preprint introduces a geolocation method, occasionally referred to as a tool, designed for application to EarthCARE. This method draws upon the experience gained from the observations and methods developed for CloudSat and CALIPSO. The geolocation method previously applied to CloudSat was based on a Digital Terrain Model (DTM) that was coarser than the ASTER DEM/WBD products used in this preprint. When applied to the CloudSat/CALIPSO datasets, the proposed method demonstrates strong performance, and the quantification of the pointing error is more precise. Examples of the application for EarthCARE geolocation are provided, though a more stringent performance indication likely requires actual data. Nevertheless, optimal areas for applying the geolocation methods within EarthCARE are defined. The manuscript is well-written and easy to understand, though some points require clarification.

We thank the reviewer for taking the time to read the manuscript and provide comments that will help improve the clarity and overall quality of our work.

**Specific Comments:**
1. Fig. 1: It would be beneficial to include a comparison with the Digital Elevation Model (without convolution) in Fig. 1c.

   The original DEM (without convolution) has been added to Fig 1c and the legend has been updated accordingly:

[Figure]

   Figure 1. [..] Panel (c) shows the original DEM (grey dashed line), the simulated DEM using the satellite's footprint (black line), the coastline location (red dashed line) and a simulated water land navigation flag (blue line). [...]

2. Line 175: Figure 3 displays the points suitable for applying EarthCARE geolocation. The selection criteria are vaguely described in the manuscript. A more detailed description of the

criteria used for selecting these points, along with their quantity, is recommended.

Thanks to the reviewer for requesting clarification on this. The criteria used to select the scenes depicted in Figure 3 are thoroughly described in the manuscript, specifically in lines 144 to 155 and 168 to 175. We believe these sections cover all necessary details regarding the selection process, ensuring that no critical information has been omitted. However, we have added clarification on the quantity of scenes selected.

"The results of the normalized overlapping areas using the distributions of the 2º×2º gridded maps shown in Fig. 2, are used to select the most suitable regions for the coastal detection. The best 100 candidates are initially selected, considering this number a solid basis for statistical analysis. After individual visual inspection, focusing on the behavior of the $\sigma_0$ measurements from the CloudSat dataset, 30 scenes are discarded."

The same process was applied in the significant elevations approach:

"Similar to the coastline analysis, 100 scenes are initially selected, with 30 discarded after visual inspection."

3. Line 281: Please provide the number of overpasses that will be available during the 3-month period.

Thanks to the reviewer for raising this point. The number of overpasses depends on the region and the configuration of the orbit. We have included this information in the text for clarity:

"While the number of overpasses depends on the region and the configuration of the orbit, initial tests using one of the simulated EarthCARE TLEs estimated that the number of monthly overpasses per scene ranged from 2 to 7."

4. Line 400: Please specify the name of the processors in the EarthCARE processing system that utilize the geolocation tool.

The processors will not directly utilize the geolocation tools described in the manuscript. Instead, the analyses presented here will be conducted using the ATLID FeatureMask (A-FM) L2a product (Zadelhoff et al., 2023) and the JAXA L1b CPR data product (called C-NOM). The geolocation results will be shared with ESA to assist in correcting the attitude data if necessary.

This information has been added to the text, more precisely in the introduction.

5. Line 404: The text mentions the application of EarthCARE co-registration, which is not described in the preprint. Is there a reference that describes the co-registration to be used in EarthCARE?

Thank you for this comment. In our manuscript, 'co-registration' refers to the alignment of datasets from different sensors, which is a critical requirement for the synergistic algorithms—this definition is provided in the introduction. There are no other references describing co-registration in EarthCARE. The methodologies presented in the manuscript are

intended for both geolocation and co-registration assessments of the active instruments. As mentioned in the summary, the co-registration between ATLID and CPR will be based on a statistical comparison of their individual geolocation assessments (line 407).